# Lipid-Lowering Treatment Gaps in Patients after Acute Myocardial Infarction: Using Global Database TriNetX

**DOI:** 10.3390/medicina60091433

**Published:** 2024-09-02

**Authors:** Grete Talviste, Mall Leinsalu, Peeter Ross, Margus Viigimaa

**Affiliations:** 1Department of Health Technologies, School of Information Technologies, Tallinn University of Technology, Ehitajate Street 5, 19086 Tallinn, Estonia; peeter.ross@taltech.ee (P.R.); margus.viigimaa@taltech.ee (M.V.); 2National Institute for Health Development, Paldiski Highway 80, 10617 Tallinn, Estonia; mall.leinsalu@tai.ee; 3North Estonia Medical Centre, J. Sütiste Road 19, 13419 Tallinn, Estonia

**Keywords:** acute myocardial infarction, LDL cholesterol, lipid-lowering treatment, cardiovascular complications, mortality, TriNetX

## Abstract

*Background and Objectives:* Patients with previous acute myocardial infarction are at significantly higher risk of recurrent events. Early and intensive lipid-lowering therapy targeting low-density lipoprotein cholesterol is a key strategy for reducing cardiovascular risk in post-acute myocardial infarction patients worldwide. This study aimed to assess patients’ real-life lipid-lowering treatment gaps after acute myocardial infarction using a global network, TriNetX, of anonymous, real-time patient data. The uniqueness of the study was the use of the novel, evolving, and constantly improving TriNetX platform and the evaluation of its feasibility for clinical research. *Materials and Methods:* A retrospective study was conducted on global repository patients in 2020, diagnosed with acute myocardial infarction, with a three-year follow-up. *Results*: After acute myocardial infarction, the prescribing rate of lipid-lowering medication (statins, ezetimibe and PCSK9I) was insufficient to reach target LDL-C values. The mean LDL-C level decreased from 2.7 mmol/L (103 mg/dL) as measured on the day of AMI to 1.97 mmol/L (76 mg/dL) between 31D and 3M. During the second and third years, the mean LDL-C value was stable (around 2.0 mmol/L (78 mg/dL)). LDL-C goals were not sufficiently reached, as only 7–12% of patients were reported to have LDL-C values < 55 mg/dL (1.4 mmol/L) and 13–20% of patients were reported to have LDL-C values < 70 mg/dL (1.8 mmol/L) during the follow-up periods. This means that a substantial number of patients remain at a very high risk for CV complications and mortality. Most cardiovascular complications happen within three months after acute myocardial infarction. *Conclusions:* Gaps remain between the recommendations for managing LDL-C in guidelines and what occurs in real life. The TriNetX platform is an innovative platform with significant potential and should be further developed for clinical research, as it enables the use of valuable interinstitutional data.

## 1. Introduction

Over the past three decades, the number of cardiovascular disease-related (CVD) deaths has risen globally, representing 20.5 million deaths in 2021. The age-standardized CVD death rate has declined by one-third worldwide, indicating progress in prevention and treatment, mainly in high-income countries. Middle- and low-income countries data suggest the opposite [1]. One of the most serious conditions and significant contributors to disability and mortality in individuals with ischaemic heart disease is acute myocardial infarction (AMI) [2]. Patients who have experienced a prior AMI face a significantly increased risk of recurrent events within the first year after the initial AMI and thereafter [3,4]. Thus, early and high-intensity lipid-lowering therapy (LLT) targeting low-density lipoprotein cholesterol (LDL-C) for patients after AMI is the primary focus of CV risk-reduction strategies worldwide [5,6,7,8]. The main challenges in managing patients with an AMI are associated with reaching target lipoprotein levels. The latest guidelines provided by the American Heart Association and American College of Cardiology (AHA/ACC) task force in managing blood cholesterol were released in 2018 and those by the European Society of Cardiology and the European Atherosclerosis Society (ESC/EAS) for the management of dyslipidaemia in 2019 [8,9].

These guidelines have several similarities in terms of primary prevention. However, there are conceptual differences in terms of secondary prevention in LDL-C target values: the European guidelines recommend an LDL-C reduction of ≥50% from the baseline and a goal of <1.4 mmol/L (<55 mg/dL) in the case of very high-risk patients, such as AMI patients [8], and the American guidelines recommend an LDL-C reduction of ≥50% from the baseline and an LDL-C of <1.8 mmol/L (<70 mg/dL) [5,9]. Despite the differences in guidelines, it has been shown that most patients at higher CV risk still do not achieve the guideline-recommended LDL-C goals [10,11,12,13]. The persistent gap between what has been suggested in the guidelines and what is happening in clinical practice regarding LLT usage and LDL-C goals needs further investigation to improve lipid management strategies.

Furthermore, considering the growing and ageing population, prevention and disease management are paramount in increasing the number of healthy life years and decreasing healthcare costs. There is a need to identify risk groups, monitor and support patient treatment journeys, and evaluate and reach treatment goals with the help of reliable data analysis, health research, and IT solutions.

With the increasing digitalization of medical records, many researchers use Electronic Health Record (EHR) data for their studies. This approach offers several benefits over traditional methods, as data gathered once are used multiple times. Leveraging EHR data allows investigators to quickly and cost-effectively access large datasets with statistical power compared to traditional data collection methods. However, EHR data have not been collected primarily for research purposes. but mainly for clinical and administrative purposes; thus, it is important to consider the complexity of EHR data usage. Nevertheless, intra- and inter-institutional patient data for research are most probably needed in the future.

Considering the above, a revolutionary and powerful global medical research platform, TriNetX, was used for this research. TriNetX represents a network of clinical data repositories that enables reliable, real-time, anonymous health data analysis for secondary purposes such as medical research. For years, there has been a gap in the market in terms of creating global clinical data repositories and a network among these organisations for secondary data use [14].

Creating such a network has proven challenging, as it involves several components, such as the development of data models and software tools for data acquisition, transformation, and storage; ensuring the interoperability of IT systems; managing data harmonization in terms of terminology, classifications, and coding systems; ensuring data security; coordination of governmental policies; and oversight. However, the main reason for the disruption of these networks has been funding. In terms of funding, TriNetX has successfully involved industrial partnerships [14,15].

This research aimed to assess LLT, LDL-C control, CV complications, and mortality after AMI worldwide, using the anonymous, real-time patient data in the TriNetX platform. The uniqueness of the study was the use of the TriNetX platform and the evaluation of its feasibility for clinical research.

## 2. Materials and Methods

The retrospective research was planned to study the TriNetX-platform global repository of patients diagnosed with AMI in 2020. LLT, LDL cholesterol control, CV complications, and mortality after AMI were analysed during several follow-up periods.

The TriNetX platform, which consists of anonymous electronic health records from 126 healthcare organizations (HCOs), involving 17 countries in North and South America, Europe, and Asia, was used.

In TriNetX, the query is a unique set of terms and logic the user sets in a specific network, which includes building the cohort and analysing the cohort in more detail in the Analytics Field. The filters for the cohort included timeframe (01.01.2020–31.12.2020), diagnostic code (ICD10 = I21), and age at the event of diagnostic code (18–70 years) and were run using the Global Collaborative Network. The query results in patient counts and is saved in the Activity History. The cohort was further analysed in the Analytics Field using Outcome Analysis, one of the TriNetX Analytics Field features.

In the Outcome Analysis, baseline characteristics were first analysed. The time window for the baseline characteristics was set (anytime up to one-day-before the index event), the index event was chosen (predefined in the cohort builder), and the features of interest were chosen (mean age at index event, sex, race, CV risk factors and diseases, laboratory measurements, measurements, lifestyle factors and LLT)—which resulted in patient counts. Next, the outcomes of different follow-up periods were analysed by setting time windows, a predefined index event and outcomes under interest (LLT, LDL-C, CV complications and all-cause mortality). Data analysis was performed on 27 June 2024.

Electronic Health Record (EHR) data are queried by TriNetX from member HCOs’ research repositories or extracted to the TriNetX environment as CSV files. Data are then mapped by TriNetX according to data exchange standards, clinical classifications and terminologies, such as Health Level 7 (HL7), the 9th revision of the International Statistical Classification of Diseases and Related Health Problems (ICD-9), the 10th revision of the International Statistical Classification of Diseases and Related Health Problems (ICD-10), the Current Procedural Terminology (CPT), standardised nomenclature for clinical drugs (RxNorm), and Logical Observation Identifiers Names and Codes (LOINC), and transformed into a specific schema involving data cleaning and quality checks. TriNetX only involves patient records with patient identifiers and at least one non-demographic fact regarding missing or partial data. However, data compactness is not guaranteed, as gaps in EHRs are not filled in as estimations by TriNetX. Rather, the TriNetX platform shares and reflects how information is entered into an EHR [15].

Demographics such as patient age, sex, and ethnicity are coded from HL7 messages. The codes for diseases and comorbidities are based on ICD-10. If an HCO has provided data in ICD-9, TriNetX uses GEMs plus custom algorithms to map and transform the data from ICD-9 to ICD-10. The Anatomical Therapeutic Chemical (ATC) classification system and the Veterans Affairs (VAs) National Formulary codes were used to identify lipid-modifying agents (ATC term) and antilipemic agents (VAs term)—group C10 according to the ATC (including plain C10A and combinations C10B) and group CV350 according to the VAs National Formulary. Laboratory tests and other clinical observations were identified using LOINC-, CPT-, RxNorm-, and TNX-curated codes.

As ESC/EAS and AHA/ACC have differences in terms of very high-risk patients’ target LDL-C, <1.4 mmol/L (<55 mg/dL) and <1.8 mmol/L (<70 mg/dL), respectively, we have included both target values in the outcome analysis.

### 2.1. Cohort

The cohort was formed of patients with AMI (ICD10 = I21), aged 18–70 at the event of AMI between 01.01.2020 and 31.12.2020 (*n* = 122,975). Patients under 18 years of age were excluded, representing single cases, and patients over 70 years of age were excluded, as they might have several co-morbidities and drugs. Thus, the analysis for lipid control might be affected for other reasons.

### 2.2. Baseline Characteristics of Interest

Age at index, sex, ethnicity, co-existing risk conditions at baseline (Hypertensive diseases ICD10 = I10–I15, Ischemic heart diseases ICD10 = I20–I25, type 2 diabetes mellitus ICD10 = E11, Dyslipidaemia ICD10 = E78, Nicotine dependence ICD10 = F17), values of lipid analysis at baseline (LDL-C, high-density lipoprotein cholesterol (HDL-C), total cholesterol (TC), triglycerides (TG), lipoprotein(a) (Lp(a)), apolipoprotein(B) (Apo(B)), and other measurements (heart rate (HR), blood pressure (BP), body mass index (BMI)).

### 2.3. Outcomes of Interest

LLT
Number of patients with LLT (ATC = C10/VA = CV350). Assessed periodically, follow-up periods: date of AMI-1 year (Y), 366 days (D)-2Y, 731D-3Y.Prescribing trends in terms of active substance. Assessed periodically, follow-up periods: date of AMI-1Y, 366D-2Y, 731D-3Y.
LDL-C control
Mean LDL-C. Assessed periodically, follow-up periods: date of AMI, 1D-1M, 31D-3M, 91D-6M, 181D-9M, 271D-1Y, 366D-2M, 731D-3M.Number of patients with LDL < 1.4 mmol/L (<55 mg/dL) and LDL ≥ 1.4 mmol/L (≥55 mg/dL). Assessed periodically, follow-up periods: date of AMI-1 year (Y), 366 days (D)-2Y, 731D-3Y.Number of patients with LDL < 1.8 mmol/L (<70 mg/dL) and LDL ≥ 1.8 mmol/L (≥70 mg/dL). Assessed periodically, follow-up periods: date of AMI-1 year (Y), 366 days (D)-2Y, 731D-3Y.
CV complications—assessed cumulatively during the follow-up periods (1M, 3M, 6M, 1Y, 2Y, 3Y). The CV complications in the current study were the following:
Subsequent ST elevation (STEMI) and non-ST elevation (NSTEMI) myocardial infarction (ICD10 = I22);Cerebral infarction (ICD10 = I63);Atherosclerosis of native arteries of the extremities (ICD10 = T70.2);Presence of cardiac and vascular implants and grafts (ICD10 = Z95).
All-cause mortality—assessed cumulatively during the follow-up periods (1M, 3M, 6M, 1Y, 2Y, and 3Y).

## 3. Results

### 3.1. Baseline Characteristics

The query was run on the Global Collaborative Network with 122 HCOs online, on 27 June. A total of 114 providers responded. The final cohort included *n* = 122,975 patients diagnosed with AMI (ICD10 = I21) between 01.01.2020 and 31.12.2020 aged 18–70 at the event of AMI. The baseline characteristics are presented in Table 1. The mean age (SD) at AMI for the study cohort was 57.6 ± 9.89 years; 64% (*n* = 79,045) were men. Most patients were White (56%). The most frequently reported CV risk factor at baseline was hypertensive diseases 46%, followed by ischemic heart diseases 46%, dyslipidaemias 45%, and type 2 diabetes mellitus 28%.

Baseline laboratory measurements include only patients with values reported in their EHR. The mean calculation includes the most recent lab values in the time window (up to one day before the index event). LDL-C measurement at baseline was reported for 54,221 (44%) patients with a mean value of 2.4 ± 1.1 mmol/L (92.7 ± 41.5 mg/dL); 45% had an HDL-C measurement reported with a mean of 1.2 ± 0.4 mmol/L (45.2 ± 17.2 mg/dL); 45% had a TC measurement reported with a mean of 4.3 ± 1.3 mmol/L (166 ± 51.3 mg/dL); and 45% had a TG measurement reported with a mean of 1.8 ± 1.7 mmol/L (156 ± 147 mg/dL). Fewer than 1% of Lp(a) and Apo(B) measurements were reported from the cohort for both variables, with mean values of 139.3 ± 172.4 nmol/L (64.8 ± 80.2 mg/dL) and 0.898 ± 0.353 g/L (89.8 ± 35.3 mg/dL), respectively.

Forty-five percent (*n* = 55,790) have at least one prescription of the LLT plain/combination (ATC = C10/VA = CV350) documented, and fifty-five percent have no record of LLT. As patients shift between drugs and drugs are used in plain and/or combination, the number of patients covered by at least one LLT prescription differs from the sum of agent-based patients. Forty-four percent of patients have a statin prescription (plain and/or in combination), 4% ezetimibe, and 1% PCSK9I at baseline (Table 1).

As shown in Figure 1, the majority (60%, *n* = 73,919) of the 122,975 patients had a record of at least one prescription of LLT (ATC = C10/VA = CV350) at the end of the first year after AMI. Considering that *n* = 10,940 patients died during the first year after AMI, the cohort decreased to *n* = 112,035 patients who were alive before the second follow-up year. Of those alive, *n* = 38,392 (34%) unique patients had a record with at least one LLT during the second follow-up year. Before the third follow-up year, *n* = 108,750 were alive. Of those alive, *n* = 32,483 (30%) individual patients had a record of at least one LLT, revealing that the prescription coverage rate declines as time passes from the AMI.

The following analysis of specific LLT agents was carried out, including plain and in combination. As patients shift between drugs and drugs are used in plain and/or in combination, the number of patients covered by at least one LLT prescription differs from the sum of agent-based patients. We found that the most preferred statin was atorvastatin, followed by rosuvastatin. Atorvastatin was prescribed for 59,410 patients during the first year after AMI, for 27,319 patients during the second year (a 24% drop in patient coverage), and for 22,131 patients during the third year (a 4% drop). Rosuvastatin in plain and in combination, was prescribed for 12,778 patients during the first year after AMI, for 7932 patients during the second year (a 3% drop), and for 7299 patients during the third year. The use of ezetimibe after AMI was identified in the case of 6394 patients during the first year, 5179 during the second year (a 19% drop), and 5142 during the third year (a 1% drop).

PCSK9I was prescribed for 1504 patients during the first year after AMI, for 1424 during the second year (a 5% drop), and for 1390 during the third year (a 2% drop). Though we did not identify any patients with inclisiran prescriptions during the first year after AMI, it was positive to see that 10 patients had inclisiran prescriptions during the second year (30 in the third year). Apart from inclisiran (the prescribing % increased), ezetimibe and PCSK9I (the prescribing % stayed roughly the same), we saw a drop in patients covered by prescriptions, as time went by, for different statins, which is associated with the overall drop in patients covered by LLT prescriptions (Table 2).

### 3.2. LDL-C Control

Data for mean LDL-C values revealed that after AMI, LDL-C decreases to 1.97 mmol/L (76 mg/dL) between 31 D and 3 M. During the second and third years, the mean LDL-C value stays in the range of 2.02–2.05 mmol/L (78–79 mg/dL) (Figure 2). However, it is also important to consider how many patients had LDL-C measured in the specific time interval. During the first year, LDL-C was measured for *n =* 54,621 patients (most measurements were carried out during the first month after AMI); during the second year, *n* = 29,450 patients; and during the third year, *n* = 26,668. The mean LDL-C values were higher than recommended for very high-risk patients, such as patients with AMI, in the 2019 ESC/EAS (<1.4 mmol/L (55 mg/dL)) and 2018 AHA/ACC guidelines (1.8 mmol/L (70 mg/dL)) (Figure 2).

A total of 12% of patients achieved LDL-C levels below 1.4 mmol/L (55 mg/dL), according to the 2019 ESC/EAS guidelines, at the end of the first-year follow-up period, 8% during the second year, and 7% during the third year. A total of 20% of patients reached LDL-C values below 1.8 mmol/L (70 mg/dL), according to the 2018 AHA/ACC guidelines, at the end of the first-year follow-up, 13% during the second year, and 13% during the third year (Table 3). As one patient might have had several measurements, the same patient might have fallen into both groups: LDL-C controlled and LDL-C uncontrolled.

### 3.3. CV Complications

The following events were most common in patients after AMI: cardiac and/or vascular implants and grafts, followed by cerebral infarction, atherosclerosis of native arteries of the extremities, and STEMI and NSTEMI myocardial infarction. During the first three months, 58% of all cardiac and/or vascular implants and grafts were performed, and 50% of all cerebral infarctions and 42% of all STEMI and NSTEMI infarctions happened. Only atherosclerosis of native arteries of the extremities increased more slowly than the other mentioned CV complications. At the end of the third-year follow-up, we identified *n* = 27,038 patients with ICD10 = Z95, *n* = 6738 patients with ICD10 = I63, *n* = 3865 patients with ICD10 = I70.2, and *n* = 880 patients with ICD10 = I22, representing 25%, 6%, 4% and 1%, respectively, from those alive at the beginning of third-year follow-up (Figure 3).

### 3.4. Mortality

Data revealed that 9% (*n* = 10,940) of the initial cohort (*n =* 122,975) died within the first year, 12% (*n* = 14,225) of patients had died by the end of the second year, and 13% (*n* = 16,354) by the end of the third year. We saw that 43% (*n* = 7060) of all deaths (*n* = 16,354) happened within three months after AMI, most within the first year (67%, *n* = 10,940) (Figure 4).

## 4. Discussion

### 4.1. Using Electronic Health-Record Data for Research

As the digitalization of medical records increases, using EHR data for research purposes is an approach many researchers have taken [16,17]; it offers several advantages over traditional study approaches, as already collected clinical data are used several times. However, EHR data have not been collected for research purposes, but primarily for clinical and administrative purposes. This means that even though more data are being digitalized and there is a continuous rise in the adoption of EHRs globally, the data gathered can be unstructured and incomplete with respect to matching data exchange standards, clinical classifications, terminologies, etc. Nevertheless, the secondary use of clinical data has promising potential for scientific research and improving health care.

Using EHR-based data in research allows investigators to access large datasets with statistical power faster and at a lower cost than traditional data gathering [18]. Moreover, EHR data represent the entire patient population, as they involves women, the elderly, and patients with several comorbidities, who are often under-represented in studies where data is collected for research purposes [19,20]. EHRs might, therefore, have a lower sample-selection bias [18]. Despite the significant advantages, it is important to address the complexity of EHR data. The main challenges in using EHR data are associated with data availability, interpretation, and missing visits and measurements [21,22]; e.g., regarding representativeness, EHR data-based studies encounter challenges due to incomplete or missing data.

We used the TriNetX live platform, which involves patient records from several HCOs’ EHRs. TriNetX includes only patient records with patient identifiers and at least one non-demographic fact. As discussed earlier, gaps in EHRs are not filled in as estimations by TriNetX; rather, it reflects how information is entered into an EHR [15].

Data availability in TriNetX was also affected by the specific technical aspects of the integration process; there were 10 HCOs out of 126 HCOs on the Global Collaborative Network that did not provide laboratory data and 7 HCOs that did not provide medication data.

### 4.2. Baseline Characteristics of AMI Patients

Baseline data reveal that most AMI patients are men (64%). Compared to earlier studies that suggest similar results [23,24,25,26,27], an extensive national inpatient study between 2006 and 2019 in the United States with over nine million AMI patients found that 61% were men [25]. However, the higher incidence rate of AMI in men might be associated with sex disparities in the diagnostics of AMI [25,28]. Moreover, several lifestyle factors and lipid abnormalities might contribute to this. Younger men tend to have a higher prevalence of lipid abnormalities and smoking [29]. Although studies suggest that the prevalence of AMI is higher in those aged > 60 years [30], current research shows a relatively low mean age (57.6 ± 9.9 years) at index event. This might be associated with the current study design, as we excluded patients > 70 years.

Our findings show that most patients with AMI were White (56%). Similarly, a study in California, including those aged ≥ 35, from 2000–2014, using discharge diagnostic codes showed that the highest incidence of AMI is among Whites [26]. US statistics between 2013 and 2021 show that the highest prevalence of AMI is among American Indians or Alaska Natives, followed by Native Hawaiians [31].

At baseline, hypertension was the most frequently reported CV risk factor (46% of patients) and has been similarly reported in several studies [25,27]. As an example, Eggers et al. investigated AMI patients between 2005 and 2017 in the SWEDEHEART Registry (Swedish Web-system for Enhancement and Development of Evidence-based Care in Heart disease Evaluated According to Recommended Therapies). They found that hypertension was present at baseline for 49% of patients. However, the cohort excluded patients who had dementia and had incomplete data for GRACE 2.0 calculation [27]. GRACE is a tool for predicting all causes of mortality in the case of acute coronary syndrome [32]. We observed 45% dyslipidaemia at baseline compared to the SWEDEHEART study with 48%. In our study, type 2 diabetes mellitus was recorded at 28% at baseline compared to the SWEDEHEART study (37%). However, we only included type 2 diabetes mellitus in the statistics, whereas SWEDEHEART used diabetes.

In terms of smoking, our results were similar to SWEDEHEART (22% vs. 21%, respectively). As the mean BMI varies widely across the globe, it is evident that the mean BMI in our study population consisting of global network patients was higher than in the SWEDEHEART registry-based study (30.6 kg/m^2^ and 26.3 kg/m^2^, respectively). Based on survey data and statistical modelling of World Health Organization (WHO) data, obesity tends to be more prevalent in wealthier countries across Europe, North America, and Oceania. For instance, in the United States, more than a third of adults were obese in 2016 [33]. In contrast, obesity rates are notably lower in South Asia and Sub-Saharan Africa.

The LDL-C mean at baseline was 2.4 mmol/L, which is lower than in the general population of the same age. However, as 46% of patients already had a diagnosis of ischemic heart disease (I20–I25), 28% type 2 diabetes mellitus (E11), and 45% dyslipidaemia (E78), they should have been targeted with much lower LDL-C values. Thus, the pre-infarction treatment was not sufficient to avoid AMI.

The mean Lp(a) value was surprisingly high at baseline (139.3 ± 172.4 nmol/L (64.8 ± 80.2 mg/dL)), adding further risk, as it is considered a causal risk factor for atherosclerotic cardiovascular disease. Although it is suggested that Lp(a) should be measured at least once in adults, and it is estimated to be elevated in one-in-five people worldwide [8,34], it was measured for only <1% of patients at baseline.

The noticeable number of patients (*n* = 67,185) with no reported LLT in their EHR at baseline and considerably lower rates of reported statin prescriptions (44%) compared to much higher rates (85%) in the Californian study [27] might be associated with data availability, as discussed in the section on EHR-based research. Nevertheless, although researchers must be aware of the complexity and limitations of EHR-based research, it offers a promising future in using existing data, improving efficiency, and giving a more realistic view of patient management [35].

### 4.3. LLT and LDL-C Control of AMI Patients

Relatively low LLT prescription coverage rates in the first, second, and third years (60%, 34%, and 30%, respectively) can be associated with the fact that seven HCOs in TriNetX are currently not providing medication data. Nevertheless, the trend in LLT coverage rate should not be affected by that, as the same HCOs provide medication data. The most preferred LLT agent prescribed after AMI was atorvastatin, which was prescribed for 48%, most likely due to its well-documented effect on lowering LDL-C [36]. The recommendation to use statins after AMI as a first-line LLT is confirmed in the guidelines: use high-dose statins as early as possible after AMI, as statins are known for their LDL-C reduction ranging from 30–50% compared to baseline values. Additionally, they may decrease triglyceride (TG) levels by 10–20% and modestly increase HDL-C levels by 1–10%. These effects contribute to better CV health. If patients do not tolerate statins or the LDL-C goal is not reached using statins alone, ezetimibe should be considered [8]. In the IMPROVE-IT study, it was shown that adding ezetimibe to statins was beneficial in terms of LDL-C reduction [37]. However, we saw ezetimibe prescriptions only for 5% of patients after AMI, a rate which stayed the same in the first-, second-, and third-year follow-up periods. Evidently, ezetimibe use was insufficient while analysing LDL-C mean values and the number of patients who reached target LDL-C values. EAS/ESC guidelines recommend adding ezetimibe 4–6 weeks after AMI if maximally tolerated statins do not bring the patient to the goal [8].

PCSK9I was used only for approximately 1% of patients during different follow-up periods. According to the guidelines, PCSK9I should be used when the combination of statins and ezetimibe does not lower LDL-C to target values. Both types of PCSK9I (alirocumab [38] and evolocumab [39]) effectively lower LDL-C levels. It has been shown that PCSK9I can reduce LDL-C by up to 60%, depending on the dosage. Furthermore, when combined with high-intensity or maximally tolerated statins, LDL-C is reduced by 46–73% more than placebo and 30% more than ezetimibe [8].

It is positive to see patients with inclisiran prescriptions in the dataset, as inclisiran is a relatively new drug, approved by the European Union in December 2020. In December 2021, it was also approved by the U.S. Food and Drug Administration (FDA). Considering that our study was followed up for three years, the end of 2023 marked the maximum period. Thus, we only saw a few prescriptions reported (10 prescriptions in the second-year and 30 prescriptions in the third-year follow-up).

While analysing the prescribing of LLT agents, in addition to following the guidelines, we must consider the rules and reimbursement (discount) of LLT agents, which vary globally. Statins are widely available in several countries with minimal or no co-payment requirements. However, there are still regions where prescribing statins—even with co-payment—is limited to specific clinical indications. The situation with ezetimibe is even more complex, as some countries still limit the allowance of prescribing rights to cardiologists and endocrinologists.

Moreover, there can be restrictions on using ezetimibe in some countries. For example, access is given only when patients are intolerant to statins. Access to PCSK9I is limited, as reimbursement rules are strict. Ezetimibe should be used before considering PCSK9I [8,9]. Compared to other LLTs (such as statins and ezetimibe), PCSK9I is much more expensive.

All in all, LDL-C goals were not sufficiently reached, as only 7–12% of patients were reported to have LDL-C values < 55 mg/dL (1.4 mmol/L) and 13–20% <70 mg/dL (1.8 mmol/L) during the follow-up periods. Thus, a substantial number of patients remain at excessive risk of CV complications and mortality.

### 4.4. CV Complications and Mortality of AMI Patients

Data revealed that 42% of all STEMI and NSTEMI myocardial infarctions, 50% of all cerebral infarctions, 58% of all cardiac and vascular implants and grafts, and 43% of all deaths happen within the first three months after AMI. This confirms what has been suggested in the guidelines: early and continuous LLT targeting LDL-C goals is of utmost importance.

### 4.5. Feasibility of TriNetX for Clinical Research

The TriNetX platform is a novel, evolving, and constantly improving platform that connects the EHR data of different HCOs worldwide. Thus, any piece of critical research that reflects the strengths and limitations of this novel platform can contribute to further data improvements and help developers better meet the needs of data users.

Regarding the feasibility of the TriNetX platform and to ensure the reliability of EHR data for clinical research, it is crucial to focus on the quality of data entry, terminology, and classifications in HCOs participating in this network.

The similarities and differences with other research are described in more detail below.

#### 4.5.1. Similarities

The data on demographics (sex, race) of patients with AMI seem to be reliable using the TriNetX platform, as other studies [23,24,25,26,27] have reported similar results. The findings from the TriNetX platform queries were consistent with other studies involving AMI patients, particularly regarding cardiovascular risk factors like hypertension [25,27], dyslipidemia [27] and smoking [27].

As discussed in ESC/EAS 2019 and AHA/ACC 2018 guidelines, the LDL-C goal attainment after AMI is low [8,9]; this was also confirmed in our study. A total of 7–12% of patients were reported to have LDL-C values < 55 mg/dL (1.4 mmol/L) and 13–20% < 70 mg/dL (1.8 mmol/L) during the follow-up periods after AMI compared to the SANTORINI study, where 80% of high- and very-high-risk patients also did not meet the LDL-C goals set by the 2019 ESC/EAS guidelines [13].

#### 4.5.2. Differences

While studies indicate that AMI prevalence is higher in individuals over 60 years old [30], recent research reveals a lower mean age at the index event. However, this discrepancy is most probably associated with the study design, as we excluded patients over 70 years old and not with the reliability of the TriNetX platform.

In our study, type 2 diabetes mellitus was recorded at a lower baseline rate than in the SWEDEHEART study. It is important to note that we only included type 2 diabetes mellitus in our statistics, whereas SWEDEHEART [27] included all types of diabetes. Thus, this difference is also most likely associated with study design and not with the feasibility of the TriNetX platform.

The mean BMI in our global study population was higher (30.6 kg/m^2^) compared to the SWEDEHEART [27] study (26.3 kg/m^2^). This difference is probably associated with population differences compared to our global (consisting mainly of HCOs from the United States) and the SWEDEHEART populations, as obesity is highly prevalent in the United States.

A relatively low rate of patients at baseline and follow-up periods whose LDL-C measurement was available and who had LLT prescriptions in the TriNetX platform can be associated with the data availability. According to TriNetX, 7 HCOs out of 126 HCOs do not provide medication data, and 10 HCOs do not provide laboratory data.

The prescribing rates of ezetimibe and PCSK9I were relatively low and were insufficient to reach LDL-C goals. However, as previously mentioned, this could be linked to the diverse populations within the global network. Factors such as reimbursement and co-payment policies, restrictions on which specialists can prescribe certain medications, and varying drug restrictions across different countries all play a role. Thus, the issue raised could be addressed in more detail if, in the future, TriNetX has the possibility of filtering data country by country.

### 4.6. Limitation

The study was conducted using EHR-based data, which have their complexities and limitations, such as data availability, interpretation, and missing visits and measurements [18,21,22,35]. According to TriNetX, 10 HCOs are not currently providing laboratory data, and seven HCOs are not providing medication data.

The participation of HCOs contributing to the TriNetX platform may be selective. There is a greater likelihood that larger centers/hospitals (e.g., university hospitals) with sufficient resources, IT capabilities and interest in the mutual sharing/use of data have joined the network.

## 5. Conclusions

Patients’ mean age at the event of AMI was relatively low (57.6 ± 9.9 years); most patients were men (64%); 56% were white; and the most common CV risk factor at baseline was hypertensive disease (46%), followed by ischemic heart diseases (46%), dyslipidaemias (45%), and type 2 diabetes mellitus (28%). LDL-C measurement before AMI was reported for 44% of patients, with a mean value of 2.4 ± 1.1 mmol/L (92.7 ± 41.5 mg/dL); and 45% had at least one prescription of LLT documented before AMI (statins were the most preferred, and the prescribing rate of ezetimibe and PCSK9I was very low).

After AMI, the prescribing rate of LLT was highest during the first year, with 60% of patients having at least one LLT prescription, which dropped afterwards (34% second year, 30% third year). The most preferred LLT agent was atorvastatin. The prescribing rates of ezetimibe and PCSK9I were relatively low, at 5% and 1%, respectively (staying in the same range for the three-year follow-up). It was positive to see that the relatively new drug inclisiran was used for some patients (*n =* 10 during the second year, *n* = 30 during the third year). The mean LDL-C level decreased from 2.7 mmol/L (103 mg/dL) as measured on the day of AMI, to 1.97 mmol/L (76 mg/dL) between 31D and 3M. During the second and third years, the mean LDL-C value was stable (2.0 mmol/L (78 mg/dL)). LDL-C goals were not sufficiently reached, as only 7–12% of patients were reported to have LDL-C values < 55 mg/dL (1.4 mmol/L) and 13–20% <70 mg/dL (1.8 mmol/L) during the follow-up periods. This means that a substantial number of patients remain at very high risk for CV complications and mortality.

Most CV complications and deaths happen within three months after AMI (42% of all STEMI and NSTEMI myocardial infarctions, 50% of all cerebral infarctions, 58% of all cardiac and vascular implants and grafts, and 43% of all deaths). Early and continuous maximally tolerated LLT-targeting LDL-C goals are, therefore, of utmost importance to avoid premature CV complications and mortality. Moreover, it is essential to address factors such as underestimation of CV risk and underutilization of combination therapy, to improve outcomes and align with current guidelines.

Even though more data are being digitalized and there is a continuous rise in the adoption of EHRs globally, the data gathered can be unstructured and incomplete with respect to matching data exchange standards, clinical classifications, terminologies, etc. Nevertheless, the secondary use of clinical data has promising potential for scientific research and improving health care. Though the network that TriNetX has created is powerful and has great potential in clinical research, as it enables the use of valuable interinstitutional data, it should be further developed.

## Figures and Tables

**Figure 1 medicina-60-01433-f001:**
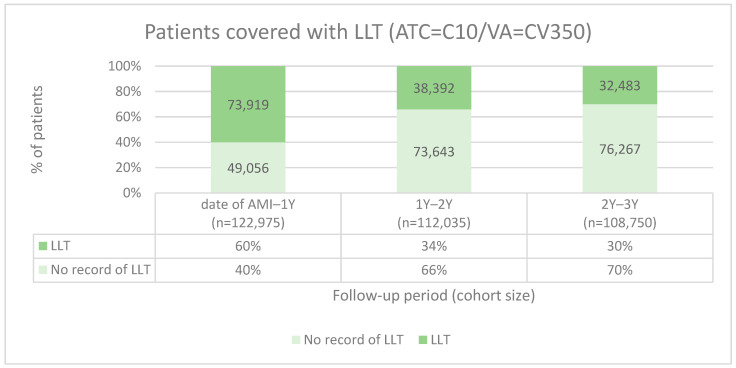
Patients covered by LLT (ATC = C10/VA = CV350) prescriptions during follow-up periods. LLT, lipid-lowering treatment; ATC, Anatomical Therapeutic Chemical Classification System; C10, lipid-modifying agents; VA, Veterans Affairs; CV350, antilipemic agents; D, day; M, month; Y, year.

**Figure 2 medicina-60-01433-f002:**
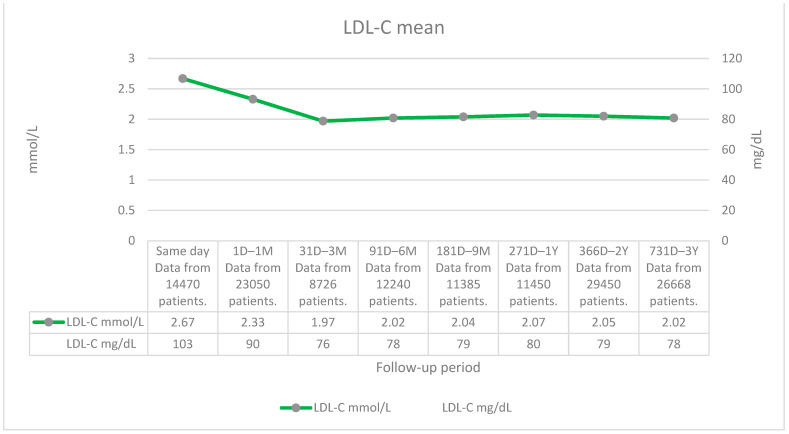
LDL-C mean (mmol/L, mg/dL). LDL-C, low-density lipoprotein cholesterol; D, day; M, month; Y, year.

**Figure 3 medicina-60-01433-f003:**
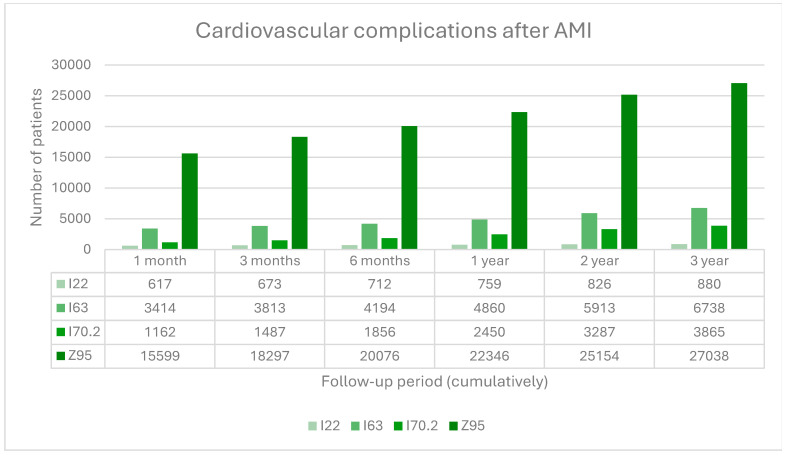
Cardiovascular complications after AMI: 122, subsequent ST elevation (STEMI) and non-ST elevation (NSTEMI) myocardial infarction; I63,cerebral infarction; I70.2, atherosclerosis of native arteries of the extremities; Z95, presence of cardiac and vascular implants and grafts.

**Figure 4 medicina-60-01433-f004:**
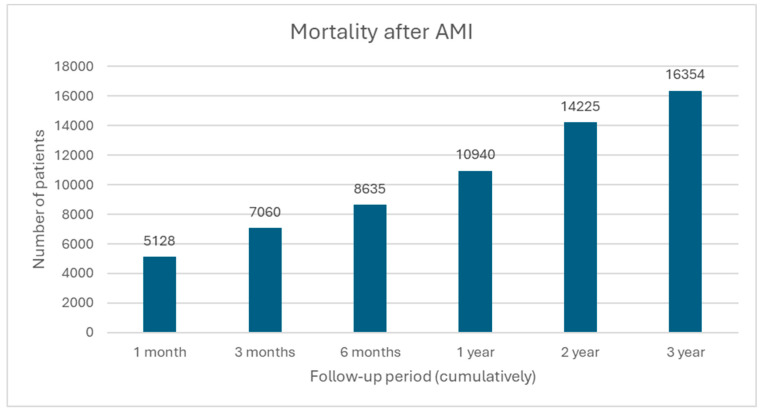
Mortality after AMI.

**Table 1 medicina-60-01433-t001:** Baseline characteristics of patients with AMI.

Characteristic	Overall *n* = 122,975
Age at index	
Mean SD	57.6 ± 9.89
Sex	
Female	39,833 (32%)
Male	79,045 (64%)
Unknown	4097 (4%)
Cardiovascular risk factors and diseases	
Hypertensive diseases (I10–I15)	64,998 (53%)
Ischemic heart diseases (I20–I25)	56,403 (46%)
Type 2 diabetes mellitus (E11)	34,804 (28%)
Dyslipidaemia (E78)	55,838 (45%)
Laboratory measurements	
LDL-C	2.4 ± 1.1 mmol/L (92.7 ± 41.5 mg/dL) Data from 54,221 (44%) patients.
HDL-C	1.2 ± 0.4 mmol/L (45.2 ± 17.2 mg/dL) Data from 54,845 (45%) patients.
TC	4.3 ± 1.3 mmol/L (166 ± 51.3 mg/dL) Data from 54,616 (44%) patients.
TG	1.8± 1.7 mmol/L (156 ± 147 mg/dL) Data from 55,096 (45%) patients.
Lp(a)	139.3 ± 172.4 nmol/L (64.8 ± 80.2 mg/dL) Data from 505 (<1%) patients.
Apo(B)	0.898 ± 0.353 g/L (89.8 ± 35.3 mg/dL) Data from 332 (<1%) patients.
Measurements	
Systolic BP	133 ± 22.5 mm/Hg Data from 63,378 (52%) patients.
Diastolic BP	77.7 ± 13.5 mm/Hg Data from 63,330 (51%) patients.
HR	78.4 ± 16.2/min Data from 53,222 (43%) patients.
Lifestyle factors	
BMI	30.6 ± 7.6 kg/m^2^ Data from 47,115 (38%) patients.
Smoking (F17)	27,276 (22%)
LLT (ATC = C10/VA = CV350)	55,790 (45%)
Statin (plain/combination)	54,523 (44%)
Ezetimibe (plain/combination)	5300 (4%)
PCSK9 (plain/combination)	1188 (1%)

LDL-C, low-density lipoprotein; HDL-C, high-density lipoprotein; TC, total cholesterol; TG, triglycerides; Lp(a), lipoprotein(a); Apo(B), apolipoprotein(B); BP, blood pressure; HR, heart rate; BMI, body mass index; LLT, lipid-lowering treatment; ATC, Anatomical Therapeutic Chemical Classification System; C10, lipid-modifying agents; VA, Veterans Affairs; CV350, antilipemic agents; PCSK9, Proprotein convertase subtilisin/kexin type 9.

**Table 2 medicina-60-01433-t002:** Prescribing trends in LLT after AMI.

Follow-Up Period Number of Patients Alive	Date of AMI-1Y *n* = 122,975	1Y-2Y *n* = 112,035	2Y-3Y *n* = 108,750
**Number (%) of patients with a prescription (trend)**			
**Agent (plain/combination)**			
Atorvastatin	59,410 (48%)	27,319 (24%)24% ↓	22,131 (20%)4% ↓
Simvastatin	2776 (2%)	942 (1%)1% ↓	748 (1%), no change
Rosuvastatin	12,778 (10%)	7932 (7%)3% ↓	7299 (7%), no change
Pravastatin	3804 (3%)	1492 (1%)2% ↓	1153 (1%), no change
Ezetimibe	6394 (5%)	5179 (5%), no change	5142 (5%), no change
PCSK9I	1504 (1%)	1424 (1%), no change	1390 (1%), no change
Inclisiran	0 (0%)	10 (<1%)<1% ↑	30 (<1%), no change

AMI, acute myocardial infarction; PCSK9I, Proprotein convertase subtilisin/kexin type 9 inhibitors, ↓ drop compared to previous period, ↑ rise compared to previous period.

**Table 3 medicina-60-01433-t003:** LDL-C goal attainment.

	Follow-Up Periods		
	1Y	1Y-2Y	2Y-3Y
Number of patients alive at the beginning of the period	122,975	112,035	108,750
Number of patients with LDL-C measurement	54,621	29,450	26,668
Number of patients LDL-C < 55 mg/dL (1.4 mmol/L)	14,596 (12%)	8347 (8%)	7594 (7%)
Number of patients LDL-C ≥ 55 mg/dL (1.4 mmol/L)	45,068 (37%)	23,121 (21%)	20,858 (19%)
Number of patients LDL-C < 70 mg/dL (1.8 mmol/L)	24,926 (20%)	14,896 (13%)	13,608 (13%)
Number of patients LDL-C ≥ 70 mg/dL (1.8 mmol/L)	36,895 (30%)	17,223 (15%)	15,400 (14%)

## Data Availability

The datasets used and/or analysed during the current study are available from the corresponding author upon reasonable request.

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
