# Peer review of "Lipid-Lowering Treatment Gaps in Patients after Acute Myocardial Infarction: Using Global Database TriNetX"

_medicina, 2024, doi:10.3390/medicina60091433_

Round 1
Reviewer 1 Report
Comments and Suggestions for Authors
This is a commendable attempt by the authors to assess the treatment gaps in using lipid lowering agents and its effects on outcomes in patients post acute myocardial infarction.
There are however significant limitations that make interpreting the data very difficult.
1. The query process is not clear. An international database was used to access the information - but individual providers caring for patients seemed to provide consent - which could have contributed to sampling bias.
2. Data availability appears to be a significant limitation (as alluded to by the authors). Baseline data and primary outcome was missing for more than 50% of the patient population.
3. The tables and figures (except table 3) show percentages without indicating the actual number of patients who had data available which may mislead viewers.
Author Response
Thank you very much for taking the time to review this manuscript. Please find the detailed responses below and the revisions/corrections highlighted in grey in the re-submitted files.
We have improved the design, results and conclusion as suggested by the Reviewer. The sections can be found on page 1, lines 18-19, lines 31-33, line 35; page 2, lines 69-77, lines 93-94; page 3, lines 102-116; page 5, lines 206-207; page 9, lines 282-284; page 10, lines 292-298, lines 313-316; page 13, lines 425-472; page 14, lines 478-481, lines 511-517.
The query process is not clear. An international database was used to access the information - but individual providers caring for patients seemed to provide consent - which could have contributed to sampling bias.
Answer 1 first part: Thank You for pointing this out. We have added a section in the revised article where we explain the query process in more detail. The section can be found under Materials and Methods on page 3, lines 102-116.
Answer 1 second part. We agree with the Reviewer's opinion that the participation of hospitals contributing to the TriNetX platform may be selective. There is a greater likelihood that larger centres/hospitals (e.g. university hospitals) with sufficient resources, IT capabilities and interest in the mutual sharing/use of data have joined the network, nor can it be ruled out that the treatment outcomes of the patients there are rather better than in hospitals on average. Also, given the large number of hospitals that have joined and the number of cases involved, we consider the impact of the sampling error on the conclusions to be minimal. We supplemented the Limitations section with a reference to a possible sampling bias on page 14, lines 478-481.
Explanation of the choice of the global database based on electronic health record data and why it might consist of lower sample selection bias:
As already discussed in the article, as the digitalisation of medical records increases, using EHR data for research is an approach many researchers have taken; it offers several advantages over traditional study approaches as data is used several times. However, EHR data has not been collected primarily for research purposes. Using EHR-based data in research allows investigators to access large datasets with statistical power faster and at a lower cost than traditional data gathering. Moreover, EHR data represents the entire patient population, as it involves women, the elderly, and patients with several comorbidities who are often underrepresented in studies where data is collected for research purposes. EHRs might, therefore, have a lower sample selection bias. Despite the significant advantages, it is important to address the complexity of EHR data. The main challenges in using EHR data are associated with data availability, interpretation, and missing visits and measurements“
Nevertheless, it is most probable, that in the future, we need both intra- and inter-institutional patient data for research, which has driven numerous initiatives over the years to establish institutional clinical data repositories (CDRs) and interconnect them across organizations. Creating such a network of CDRs constitutes many challenges, such as data models and software tools for data acquisition, transformation, and storage; ensuring the interoperability of IT systems; managing data harmonization in terms of terminology, classifications, and coding systems; ensuring data security; coordination of governmental policies; oversight and funding. Thus, many initiatives have not succeeded in establishing such network. In that sense, TriNetX is unique as they have shown a sustainable approach to federated CDRs over the past eight years.
To answer directly to the question, we don't think there will ever be such a network of EHRÅ› where all the hospitals worldwide are connected. There will always be a certain number of people left out; thus, such a database-based study always depends on the willingness of HCOs to join the network. However, TriNetX has been shown to increase the number of HCOÅ› connected year by year. Moreover, they have shown the capacity to develop and add new features to the system constantly.
In conclusion, we know the limitations of EHR data-based research compared with randomised control trials. However, we were eager to test and show that the constantly evolving TriNetX network can be used for medical research.
Therefore, according to our answer, we added some parts to the revised article to clarify. These changes can be found on Page 1, lines 18-19; page 2, lines 69-77 and 93-94; and Page 10, lines 292-298.
Comment 2: Data availability appears to be a significant limitation (as alluded to by the authors). Baseline data and primary outcome was missing for more than 50% of the patient population.
Answer 2: We totally agree with Reviewer, and as already discussed in the article, data availability is one of the main problems in all the research done using EHR data. The reasons associated with data availability are mainly because medical data is not gathered for research purposes but primarily for clinical and administrative purposes. This means that even though more data is being digitalised and there is a continuous rise in the adoption of EHRs globally, the data gathered can be unstructured and incomplete to match data exchange standards, clinical classifications, terminologies, etc. Nevertheless, the secondary use of clinical data has promising potential for scientific research and improving health care as the data is already collected. Thus, there is no need for physical intervention with patients, data can be collected quickly with a large sample size, costs for gathering data are considered low, and data collected in real-life settings are highly generalized, enhancing representative sampling and boosting validity.
In TriNetX at baseline, data was available for mean age at index, sex and race for all patients. As mentioned in the article, there were 10 HCOs out of 126 HCOÅ› on the Global Collaborative Network that do not provide laboratory data and 7 HCOs that do not provide medication data.
We have made some changes that also overlap with the previous comment changes. Changes can be found on page 2, lines 69-77 and page 10, lines 313-316.
Comment 3: The tables and figures (except table 3) show percentages without indicating the actual number of patients who had data available which may mislead viewers.
Answer 3: We thank the Reviewer for this valuable comment. We have now corrected Table 1 on page 5 and Figure 2 on page 8, as requested.
Figure 1 already has the total patient count in brackets, just after the follow-up period. Table 2. also had the total patient count; it is on the second line and marked as „Number of patients alive“.
Figure 3 and Figure 4 show the cumulative numbers of CV complications and deaths in the whole cohort, with an initial number of patients shown in the text.
Reviewer 2 Report
Comments and Suggestions for Authors
The paper “Lipid-lowering treatment gaps in patients after acute myocardial 2 infarction: using global database TriNetX” by Talviste G, et al. is a retrospective study conducted on a global repository of patients diagnosed in 2020 with acute myocardial infarction followed-up for 3 years aiming to assess patient real-life lipid-lowering treatment gaps. The repository, TriNetX, included anonymous real-time patient data.
The authors have found that the mean LDL-C level decreased from 2.7 mmol/L (103 mg/dL) as measured on the day of AMI to 1.97 mmol/L (76 mg/dL) 22 between 31D and 3M. During the second and third years, the mean LDL-C value was stable (around 2.0 mmol/L (78 mg/dL)). LDL-C goals were not sufficiently reached as only 7-12% of patients were reported to have LDL-C values <55 mg/dL (1.4 mmol/L) and 13-20% of patients were reported to have LDL-C values <70 mg/dL (1.8 mmol/L) during the follow-up periods. The authors concluded that gaps remain between the recommendations for managing LDL-C in guidelines and what occurs in real life. Hence, a substantial number of patients remain at a very high risk for CV complications and mortality.
Tables and figures: They are fine. My only comment is with regard to Figure 2. The line depicting LDL in mg/dL and mmol/L are identical since these values differ by a constant (X40). It would clearer if the figure would contain one line with the different dimensions on both sides of the figure (as in the present form) or possibly on the left. This is a minor style issue.
The paper is based on a large database and the message is clear. The idea to analyze a large repository is a good initiative. However, the results are not very novel and similar studies and conclusions have been published.
Comments on the Quality of English LanguageVery reasonabe, minor editing
Author Response
We appreciate Reviewers time and effort in reviewing our manuscript. Below, we provide detailed responses to the comments. The revisions and corrections are highlighted in grey in the re-submitted documents. Comment 1: Tables and figures: They are fine. My only comment is with regard to Figure 2. The line depicting LDL in mg/dL and mmol/L are identical since these values differ by a constant (X40). It would clearer if the figure would contain one line with the different dimensions on both sides of the figure (as in the present form) or possibly on the left. This is a minor style issue. Answer 1: We agree with the comment and have corrected Figure 2 on page 8. Comment 2: The paper is based on a large database and the message is clear. The idea to analyze a large repository is a good initiative. However, the results are not very novel and similar studies and conclusions have been published. Answer 2: Thank You for your very valuable comment. With the increasing digitalisation of medical records, many researchers use Electronic Health Records (EHR) data for their studies. This approach offers several benefits over traditional methods, as data gathered once is used multiple times. Leveraging EHR data allows investigators to quickly and cost-effectively access large datasets with statistical power compared to traditional data collection methods. However, EHR data has not been collected primarily for research purposes but mainly for clinical and administrative purposes; thus, it is important to consider the complexity of EHR data usage. Nevertheless, intra- and inter-institutional patient data for research is most probably needed in the future. Considering the above, we were eager to use the TriNetX novel platform. The study's uniqueness was the use of a novel, evolving, and constantly improving TriNetX platform, as well as clinical results and conclusions and evaluation platform feasibility for clinical research. The difference between our study and traditional studies lies in using global real-life data, as similar research is often done using national registry-based data and/or is limited by the data set size. Similar results and conclusions published before are giving us confidence in the feasibility and trustfulness of the platform. To highlight the uniqueness of the study, we have revised the manuscript and made corrections: Pages 1, lines 18-19, lines 31-33; page 2, lines 69-77, lines 93-94; page 13, lines 425-434; page 14, lines 511-517.
Reviewer 3 Report
Comments and Suggestions for Authors
Statin therapy in patients after myocardial infarction is essential. Initially, high-dose statin therapy is usually used.
In the paper initially very low rate of lipid-lowering treatment - 45%, why?
Can the authors explain the low frequency of statin prescriptions during follow-up?
Are the doses of statins known?
It is also important all other treatments. Were patients taking other treatments?
Were they patients with and without STEMI?
Did the patients undergo interventional interventions, what percentage?
Simvastatin is generally not used today in patients with AMI, because of myopathy we cannot give a high dose. Such patients should be excluded from the analysis.
It would be helpful to indicate by country what percentage of statins are prescribed!
It would be desirable to analyze and compare patients who did and did not take statins.
Author Response
We thank Reviewer 2 for taking the time to review this manuscript. Our detailed responses to comments are presented below. The revisions/corrections are highlighted in grey in the re-submitted files.
We have improved the design, method, results and conclusion as suggested by the Reviewer. The sections can be found on page 1, lines 18-19, lines 31-33, line 35; page 2, lines 69-77, lines 93-94; page 3, lines 102-116; page 5, lines 206-207; page 9, lines 282-284; page 10, lines 292-298, lines 313-316; page 13, lines 425-472; page 14, lines 478-481, lines 511-517.
Comment 1: In the paper initially very low rate of lipid-lowering treatment - 45%, why?
Answer 1: We believe that part of the explanation for 45% of patients using LLT at baseline relies on data availability. We recognize the problem of data availability when explaining the low rate of LLT. Nevertheless, we assume that, to a great extent, these numbers still reflect insufficient treatment, thus not affecting our main conclusions in any major way.
Firstly, as mentioned in the article, according to TriNetX, 7 HCOs out of 126 HCOs do not provide medication on the Global Collaborative Network due to an ongoing specific technical aspect of the integration process.
Secondly, data availability is one of the main problems in all the research done using EHR data. The article addresses the reasons besides integration that can be associated with the fact that clinically gathered data is reused (page 10, lines 290-292 and lines 299-308); in other words, medical data is not gathered for research purposes but primarily for clinical and administrative purposes. This means that even though more and more data is being digitalised and there is a continuous rise in the adoption of EHRs globally, the data gathered can be in an unstructured format, incomplete to match with data exchange standards, clinical classifications and terminologies etc. Nevertheless, the secondary use of clinical data has promising potential for scientific research and improving health care as the data has already been collected. Thus, there is no need for physical intervention with patients, data can be collected quickly with a large sample size, costs for gathering data are considered low, and data collected in real-life settings are highly generalized, enhancing representative sampling and boosting validity.
All in all, it is very likely that in the future, we need both intra- and inter-institutional patient data for research, which has driven numerous initiatives over the years to establish institutional clinical data repositories (CDRs) and interconnect them across organizations. Creating such a network of CDRs constitutes many challenges, such as data models and software tools for data acquisition, transformation, and storage; ensuring the interoperability of IT systems; managing data harmonization in terms of terminology, classifications, and coding systems; ensuring data security; coordination of governmental policies; oversight and funding. Thus, many initiatives have not succeeded in establishing such databases. In that sense, TriNetX is unique as they have shown a sustainable approach to federated CDRs over the past eight years.
TriNetX has been shown to increase the number of HCOÅ› connected year by year. Moreover, they have shown the capacity to develop and add new features to the system constantly.
In conclusion, we know the limitations of EHR data-based research compared with randomised control trials. However, we were eager to test and show that the constantly evolving TriNetX network can be used for medical research.
We have modified the article on page 2, lines 69-77 and page 10, lines 292-298 and lines 313-316.
Comment 2: Can the authors explain the low frequency of statin prescriptions during follow-up?
Answer 2: The frequency of statin prescriptions in the follow-up periods can also be connected to the data availability as described in the first answer. The section was also described in the article under Discussion, 4.3 LLT and LDL-C control of AMI patients page 12, lines 371-374.
Comment 3: Are the doses of statins known?
Answer 3: Unfortunately, data about medication doses is currently unavailable in the TriNetX database.
Comment 4: It is also important all other treatments. Were patients taking other treatments?
Answer 4: We agree with the Reviewer that other medications are also important. However, we wanted to keep the focus of this article on lipid-modifying agents. Thus, we did not include standard medication such as beta-blockers, ACE inhibitors and antiplatelet drugs in our analytical framework.
Comment 5: Were they patients with and without STEMI?
Answer 5: The cohort was designed using ICD10 code I21 Acute myocardial infarction, including both patients with STEMI and NSTEMI.
Comment 6: Did the patients undergo interventional interventions, what percentage?
Answer 6: Our study specifically aimed to assess patients’ real-life lipid-lowering gaps after acute myocardial infarction and rather focused on showing that there is great potential and future in medical research using a global network, TriNetX, of anonymous, real-time patient data. Thus, we did not specifically analyse coronary interventions; first, we deliberately tried to keep queries as simple as possible, and secondly, we concentrated the work on lipid-modifying treatment after AMI. Especially knowing that despite having coronary interventions, lipid-lowering drugs should still be used. However, we did address the whole ICD10 code Z95 Presence of cardiac and vascular implants and grafts as a complication (also involving coronary interventions) after AMI. We analysed Z95 during the different follow-up periods.
Comment 7: Simvastatin is generally not used today in patients with AMI, because of myopathy we cannot give a high dose. Such patients should be excluded from the analysis.
Answer 7: We are aware of the high myopathy risk in patients treated with simvastatin. However, as our study reflects the real-life situation, according to the data, simvastatin was still used for 2% of patients. We believe that it is important to also show the numbers reflecting simvastatin use. It is a favourable trend of decline in simvastatin use over time.
Comment 8: It would be helpful to indicate by country what percentage of statins are prescribed!
Answer 8: We were also eager to compare countries; however, it is not currently possible to filter countries within the network the North-Estonia Medical Centre can use in the TriNetX platform. We have added this suggestion also in the article on page 13, lines 470-472.
Comment 9: It would be desirable to analyze and compare patients who did and did not take statins.
Answer 9: Initially, we also tried to design cohorts as the Reviewer suggested. However, as we were dealing with real-life anonymised data, it was not possible to control medication adherence and persistence in time. Setting the filters at baseline (patients with ATC=C10 vs patients without ATC=C10) queries did not mean that patients stayed in their group. In real life, patients shift between groups.
It is important to understand that the TriNetX platform is a novel, evolving and constantly improving platform connecting EHR data in different HCOÅ› worldwide. Thus, any piece of critical research that reflects the strengths and limitations of this novel platform can contribute to further data improvements and help developers better meet the needs of data users.
Round 2
Reviewer 3 Report
Comments and Suggestions for Authors
No